# Intraocular Pressure-Lowering and Retina-Protective Effects of Exosome-Rich Conditioned Media from Human Amniotic Membrane Stem Cells in a Rat Model of Glaucoma

**DOI:** 10.3390/ijms24098073

**Published:** 2023-04-29

**Authors:** Hye-Rim Seong, Chan Ho Noh, Sangryong Park, Sumin Cho, Seok-Jin Hong, Ah-young Lee, Dongho Geum, Soon-Cheol Hong, Dongsun Park, Tae Myoung Kim, Ehn-Kyoung Choi, Yun-Bae Kim

**Affiliations:** 1College of Veterinary Medicine, Chungbuk National University, Cheongju 28644, Republic of Korea; 2Central Research Institute, Designed Cells Co., Ltd., Cheongju 28576, Republic of Korea; 3Department of Biomedical Science, Korea University College of Medicine, Seoul 02841, Republic of Korea; 4Department of Obstetrics and Gynecology, Korea University College of Medicine, Seoul 02841, Republic of Korea; 5Department of Biology Education, Korea National University of Education, Cheongju 28173, Republic of Korea

**Keywords:** glaucoma, human amniotic membrane stem cell, exosome-rich conditioned medium, intraocular pressure, retinal cell protection

## Abstract

Glaucoma is one of the most devastating eye diseases, since the disease can develop into blindness and no effective therapeutics are available. Although the exact mechanisms and causes of glaucoma are unknown, increased intraocular pressure (IOP) has been demonstrated to be an important risk factor. Exosomes are lipid nanoparticles secreted from functional cells, including stem cells, and have been found to contain diverse functional molecules that control body function, inhibit inflammation, protect and regenerate cells, and restore damaged tissues. In the present study, exosome-rich conditioned media (ERCMs) were attained via hypoxic culture (2% O_2_) of human amniotic membrane mesenchymal stem cells (AMMSCs) and amniotic membrane epithelial stem cells (AMESCs) containing 50 times more exosome particles than normoxic culture (20% O_2_) medium (NCM). The exosome particles in ERCM were confirmed to be 77 nm in mean size and contain much greater amounts of growth factors (GFs) and neurotrophic factors (NFs) than those in NCM. The glaucoma-therapeutic effects of ERCMs were assessed in retinal cells and a hypertonic (1.8 M) saline-induced high-IOP animal model. CM-DiI-labeled AMMSC exosomes were found to readily penetrate the normal and H_2_O_2_-damaged retinal ganglion cells (RGCs), and AMMSC-ERCM not only facilitated retinal pigment epithelial cell (RPEC) proliferation but also protected against H_2_O_2_- and hypoxia-induced RPEC insults. The IOP of rats challenged with 1.8 M saline increased twice the normal IOP (12–17 mmHg) in a week. However, intravitreal injection of AMMSC-ERCM or AMESC-ERCM (3.9–4.5 × 10^8^ exosomes in 10 μL/eye) markedly recovered the IOP to normal level in 2 weeks, similar to the effect achieved with platelet-derived growth factor-AB (PDGF-AB, 1.5 μg), a reference material. In addition, AMMSC-ERCM, AMESC-ERCM, and PDGF-AB significantly reversed the shrinkage of retinal layers, preserved RGCs, and prevented neural injury in the glaucoma eyes. It was confirmed that stem cell ERCMs containing large numbers of functional molecules such as GFs and NFs improved glaucoma by protecting retinal cells against oxidative and hypoxic injuries in vitro and by recovering IOP and retinal degeneration in vivo. Therefore, it is suggested that stem cell ERCMs could be a promising candidate for the therapy of glaucoma.

## 1. Introduction

Glaucoma is one of the leading causes of blindness. It is estimated that there will be 111.8 million glaucoma patients by 2040 [1,2,3]. Glaucoma is an incurable neurodegenerative disease characterized by selective, progressive, and irreversible degeneration of retinal cells, including retinal ganglion cells (RGCs) and optic nerves, following increased intraocular pressure (IOP) [4]. Both structural and functional damage to RGCs and their axons result in loss of nerve and axillary structures, activation of neuroglial cells, tissue remodeling, and change in eye blood flow [5]. Glaucoma patients without proper treatment may experience progressive narrowing of the aqueous humor drainage channel, pain from increased IOP, and, finally, blindness due to loss of neural cells and atrophy of optic nerve fibers [6].

However, the exact mechanisms of pathological changes or axonal damage due to glaucoma are not yet elucidated. Although increased IOP has been suggested to be a major cause of glaucoma, it is now known that high IOP is not the only pathological factor. There are other factors believed to be involved in the development of glaucomatous neuropathy, such as inflammatory processes, oxidative stresses, metabolic abnormalities, and blood flow disturbances [7]. Actually, a broad spectrum of causative agents is involved in the induction of glaucoma and retinal degeneration, including increased IOP, hypoxic insult, oxidative stress, excitotoxicity, activated Müller microglia-derived inflammatory cytokines, lack of neurotrophins such as brain-derived neurotrophic factor (BDNF), and so on [8].

Currently, there are three main treatments for glaucoma: medicinal, laser, and surgical therapies. Most of the therapeutics are eye drops for decreasing IOP, typically prostaglandin analogs, which are aqueous humor spill promoters and β-blockers that decrease aqueous humor inflow [7]. However, eye drops must be used lifelong every day, and since they are palliative, it is difficult to prevent glaucoma. In addition, adverse effects such as conjunctivitis, dry eye syndrome, and itching accompany. The currently employed therapeutic options are not sufficient to prevent vision loss in glaucoma patients [9,10].

More advanced attempts have been made to directly prevent the death of retinal cells through nerve protection mechanisms instead of indirect methods of lowering IOP. In particular, unlike the Western patients with high-pressure glaucoma, about 90% of Asian patients have normal IOP. In contemplating a systematic approach to neuroprotection, the main areas to target include (1) neurotoxic agents such as nitric oxide (NO) and glutamate, (2) deprivation of internal neurotrophins, (3) balancing self-repair with self-destruction in ocular nerve tissue, and (4) ocular blood flow and combating ischemia [11]. Indeed, various kinds of neuroprotective agents have been attempted for the attenuation or recovery of degenerative retinal disorders. Those include natural products, antioxidants, neuropeptides, autophagy inducers, complement inhibitors, etc. [8]. However, to date, highly-effective therapeutic materials in clinical trials are not available.

Recently, stem cells and stem-cell-derived active ingredients have been widely studied for the treatment of systemic and local diseases. Notably, positive research results on corneal damage, retinopathy, and macular degeneration are also presented in the case of eye diseases [12]. However, research on differentiating human stem cells into RGCs and transplanting them into the retina has not made progress, and the amount of active ingredients derived from simply cultured stem cells is very small, and the water-soluble protein-based cytoprotective and regenerating factors are difficult to reach the damaged area [13]. By comparison, in experimental glaucoma models, intraocular injection of stem cells or their conditioned media (CMs) collected from hypoxic culture were more effective than normally-cultured conditioned media (NCMs) in both lowering the IOP and protecting the RGCs [14]. Notably, it was proposed that hypoxia-cultivated mesenchymal stem cells (MSCs) induced trabecular meshwork regeneration for the drainage of ocular fluid [13]. More interestingly, growth factors (GFs) and neurotrophic factors (NFs) including platelet-derived growth factor (PDGF) and BDNF are proposed as main ingredients for the paracrine effects of hypoxia-cultivated stem cells [15,16,17] and are known to promote nerve survival and control synapses in glaucoma eyes [18]. In addition, stem cells and stem cell-derived vascular endothelial growth factor (VEGF) and insulin-like growth factor (IGF) exert antioxidative activities which are expected to protect retinal cells against oxidative injury [19,20,21].

It has been confirmed that functional molecules are released in extracellular vesicles (EVs) from functional cells including inflammatory and stem cells. Therefore, efforts to utilize EVs containing more than hundreds to thousands of active ingredients are bearing fruit. EVs are nanoparticles of lipid-membrane structures secreted from the functional cells as a means of encapsulating and delivering functional substances such as intercellular signal transmitters, cytokines, and GFs/NFs. EVs containing the functional molecules have been shown to regulate bodily functions, enhance or inhibit inflammation, protect and regenerate cells, and repair tissues [22,23].

In particular, among EVs from stem cells, exosomes with a size of 50–200 nm can easily reach anywhere in the damaged area, block inflammation, protect tissues, and are used as materials to regenerate cells and tissues [24,25]. However, the number of exosomes secreted from stem cells is not enough to use for disease therapy, and if the exosomes burst during the extraction process, it is difficult for water-soluble proteins to pass through the biological lipid membrane barriers.

In our studies, exosome-rich conditioned media (ERCMs) were collected from human amniotic membrane mesenchymal stem cells (AMMSCs) and amniotic membrane epithelial stem cells (AMESCs) via a hypoxic-culture process. Actually, ERCMs containing a large number of exosomes, 50 to 100 times the NCM, were obtained through hypoxic (2–3% O_2_) culture after specific cell-activation process treating with tumor-necrosis factor-α (TNF-α, 50 nM) [26,27]. In our recent study, the ERCMs restored diabetic retinopathy (manuscript in revision).

In the present study, the active ingredients in ERCMs and cell-penetrating potential of their exosomes were analyzed, and the protective effects against oxidative stress- and hypoxia-induced insults were evaluated in retinal pigment epithelial cells (RPECs). IOP-lowering and retinal-cell- and tissue-protective activities of ERCMs were confirmed to verify glaucoma-therapeutic potentials in an experimental animal model of glaucoma in comparison with NCM and PDGF-AB, a reference material.

## 2. Results

### 2.1. Characteristics of Exosomes in NCMs and ERCMs

The contents and numbers of exosomes in NCMs and ERCMs were measured via CD9 markers and nanoparticle tracking analysis (NTA), respectively. The contents of CD9-positive exosomes were very low in AMMSC-NCM and AMESC-NCM obtained in a normoxic condition (20% O_2_). By comparison, the CD9 contents in AMMSC-ERCM and AMESC-ERCM collected after a hypoxic culture (2% O_2_) were much higher (Figure 1A & Appendix A).

From NTA, the particle numbers of exosomes in AMMSC-NCM and AMESC-NCM were 8.8 × 10^8^/mL and 8.2 × 10^8^/mL, respectively. However, the exosome numbers in AMMSC-ERCM and AMESC-ERCM were 4.5 × 10^10^/mL 3.9 × 10^10^/mL, respectively (Figure 1B). Therefore, it was confirmed that the exosomes in ERCMs increased up to 50 times the NCMs.

The mean size of exosomes in the major peak from AMMSCs were found to be 77 nm (Figure 1C), although several small peaks were found which might be due to aggregation of the exosomes.

In ELISA, the concentrations of fibroblast growth factor (FGF), epidermal growth factor (EGF), IGF, VEGF, transforming growth factor-β (TGF-β), BDNF, and PDGF, that is, GFs and NFs related to retinal cell protection and regeneration, were very low in normal AMMSC-NCM (normoxic culture). By comparison, it was confirmed that the GFs and NFs were very high in AMMSC-ERCM (hypoxic culture), reaching ten to a hundred times the AMMSC-NCM (Figure 1D).

Notably, it was confirmed that the CM-DiI-labeled AMMSC exosomes readily penetrate both the normal and H_2_O_2_-treated primary RGCs, although a higher number of exosomes was found in the damaged cells (Figure 1E).

### 2.2. Retinal-Cell-Proliferative and -Protective Activities of ERCM

The proliferative activity of ERCM was confirmed in ARPE-19 cells (Figure 2A). In normoxic conditions (20% O_2_), AMMSC-ERCM facilitated the proliferation of the cells in a concentration-dependent manner. Especially at high concentrations (25–100 μL/mL), a significant increase in the ARPE-19 cell proliferation was observed.

As an oxidative stress, ARPE-19 cells exposed to 200 μM H_2_O_2_ for 24 h resulted in 42% death (Figure 2B). However, simultaneous treatment with AMMSC-ERCM rescued the retinal cells in a concentration-dependent manner. Notably, AMMSC-ERCM at high concentrations (25–100 μg/mL) fully overcame the oxidative injury of the cells and, furthermore, facilitated their proliferation more than the number of normal (uninjured) cells, up to 138%.

The cytoprotective ability of ERCM was also evaluated under a hypoxic insult. ARPE-19 cells were treated with AMMSC-ERCM and cultivated for 24 h in a hypoxic (2% O_2_) condition. Such hypoxic conditions led to a loss of the cells by 39% (Figure 2C). However, treatment with AMMSC-ERCM at high concentrations (≥25 μg/mL) fully reversed the cell death and, furthermore, facilitated their proliferation more than the number of normal (uninjured) cells, up to 185% (Figure 2D).

### 2.3. IOP-Lowering and Retina-Protective Activities in Glaucoma Animals

In a hyperosmotic glaucoma model, administration of hypertonic (1.8 M) saline caused high IOP within 7 days, which lasted longer than 3 weeks (Figure 3A). Notably, the increased IOP was recovered to normal level (10–17 mmHg) in 10 days following intravitreal injection of AMMSC-ERCM or AMESC-ERCM, comparable with the effect of PDGF-AB. In comparison, a partial recovery was achieved with AMMSC-NCM.

After induction of hyperosmotic glaucoma, the features of hematoxylin/eosin-stained retina exhibited overall shrinkage (Figure 3B). Indeed, the thicknesses of the entire retinal layer displayed severe atrophy by 35%. However, the retinal tissues were nearly fully preserved by treatment with AMMSC-ERCM, AMESC-ERCM, or PDGF-AB, but AMMSC-NCM exerted a partial effect (Figure 3C).

In glaucoma eyes, there was severe shrinkage of RGC layer and loss of RGCs in the layer, the most-vulnerable tissue in glaucoma patients (Figure 3D,E). Notably, such ganglion cell layer atrophy was near-fully recovered by treatment with AMMSC-ERCM or AMESC-ERCM, similar to the effect of PDGF-AB. In addition, the RGCs were markedly preserved by the ERCMs and PDGF-AB. However, AMMSC-NCM was much less effective in the prevention of both cell death and tissue degeneration.

### 2.4. RGC-Preserving and Neuroprotective Activities of ERCM

From the immunostaining to confirm the RGC-protective activity of NCM and ERCMs, the number of NeuN-positive RGCs markedly decreased in glaucoma rats to half the level of normal animals (Figure 4A,B). Such a loss of RGCs was nearly fully protected by the injection of AMMSC-ERCM, AMESC-ERCM, or PDGF-AB, wherein their effects were superior to AMMSC-NCM.

Since the Müller cell activation represents tissue injuries including optic verves, the immunoreactivity on GFAP, the component of Müller cells was examined. From the immunostaining, increased GFAP expression was identified in the glaucoma retina (Figure 4C,D). In the normal retina, GFAP was normally expressed in the ganglion cell layer and the nerve fiber layer, whereas in the glaucomatous retina, GFAP was over-expressed in Müller cells in damaged retinas and in the inner layers. Interestingly, however, such increased GFAP expression was markedly attenuated by the treatment of AMMSC-ERCM, AMESC-ERCM, or PDGF-AB compared to a negligible effect of AMMSC-NCM.

As another marker of Müller cell activation, GS was also confirmed to be highly expressed in the inner and outer layers of glaucoma retinas (Figure 4E,F). In parallel with GFAP, the increase in GS level was fully prevented by the injection of AMMSC-ERCM or AMESC-ERCM. PDGF-AB also showed a significant activity compared to a mild effect of AMMSC-NCM.

## 3. Discussion

In the present study, we successfully collected ERCMs containing a large number of exosomes and high concentrations of GFs and NFs by stimulating AMSCs with TNF-α and hypoxic culture. The ERCMs protected retinal cells against oxidative and hypoxic insults in vitro and restored IOP and retinal degeneration in vivo.

As a safety issue, although autologous stem cells may not induce immune rejection and toxicity, it was demonstrated that AMSCs did not cause immune rejection and inflammation, too, since they do not express major histocompatibility class (MHC) type II [28]. In addition, the ERCMs obtained AMMSCs and AMESCs were confirmed safe in non-clinical toxicity tests, including single-dose toxicity, repeated-dose toxicity, antigenicity, and safety pharmacology (unpublished data). So, we selected amniotic membranes as a source of stem cells to avoid immune adverse effects.

It is of interest that the particle size of AMMSCs (mean 77 nm) was much smaller (Figure 1C) than exosomes from other mesenchymal stem cells such as adipose-derived mesenchymal stem cells (ADMSCs; mean 220 nm) and umbilical cord blood-derived mesenchymal stem cells (UCBMSCs; mean 120 nm) [29]. Such a small size may be advantageous for the penetration of body membranes such as blood–retinal barrier (BRB), blood–brain barrier (BBB), and the skin. Actually, it was confirmed that the CM-DiI-labeled exosomes readily penetrate both the intact and H_2_O_2_-injured pRGCs (Figure 1E).

The rationale of the exosomes’ usability might be based on the functional molecules in them including GFs and NFs, exerting cell-proliferative and tissue-protective activities. In AMMSC-ERCM, high concentrations of GFs and NFs, that is, ten to a hundred times the AMMSC-NCM, were found (Figure 1D). Accordingly, ERCMs greatly enhanced the pRGC proliferation and protected ARPE-19 cells against oxidative and hypoxic injuries (Figure 2). Among GFs and NFs, PDGF was proposed as one of the main ingredients for the retina-protective and -restorative efficacies [15,16,18]. In this study, we injected 1.5 μg PDGF-AB to achieve excellent recovery of IOP and retinal injuries. Since the concentration of PDGF in AMMSC-ERCM was 205.3 pg/mL, only 2.05 pg/10 μL/eye was injected. In spite of the very small amount of PDGF in ERCM, its efficacy was comparable with the pure PDGF-AB. However, it is well known that stem cell exosomes contain thousands of functional molecules including GFs, NFs, miRNAs, cytokines, and other proteins [30]. In fact, ERCMs containing larger amounts of GFs and NFs showed much higher effects than NCM. Therefore, it is believed that various kinds of functional molecules exerted combinational and synergistic effects in cytoprotection, cell proliferation, tissue regeneration, anti-inflammation, immune modulation, and so on.

Noteworthy, the increased IOPs in hyperosmotic glaucoma model were rapidly decreased by ERCMs or PDGF-AB to a normal level in 10 days. It was proposed that hypoxia-cultivated MSCs induced trabecular meshwork regeneration for the drainage of ocular fluid [14], in which PDGF was one of the active molecules for the activity [15,16,18]. In the present study, PDGF-AB displayed a similar effect (Figure 3A). Notably, such beneficial effects were also obtained with ERCMs containing PDGF and other GFs and NFs.

Most importantly, ERCMs effectively protected against cell loss and tissue shrinkage in glaucoma eyes. It was inferred that such effects were from the retinal cell-protective activity in vitro (Figure 2A). Indeed, AMMSC-ERCM (≥25 μg/mL) fully overcame both the oxidative and hypoxic damages of ARPE-19 cells and further increased the cell numbers more than normal (uninjured) cells (Figure 2B–D). Currently, the most important pathophysiological feature in glaucoma is the selective extinction of RGCs [31]. Indeed, ERCMs and PAGF-AB completely preserved RGCs and retinal thickness at normal levels and features as observed in hematoxylin/eosin-stained tissue sections (Figure 3B–E). It is also believed that IOP decreases in ERCM- or PAGF-AB-treated eyes might be due to the recovery of retinal shrinkage and preservation of trabecular meshwork via elimination of intraocular coagulation.

In order to demonstrate more clearly the neuroprotective activity of ERCMs, we immunostained the retinal tissues with a NeuN antibody. Since NeuN is expressed in neuronal RGCs, its loss is a hallmark of retinal neuropathy [32,33], as seen in ocular diseases that reduce visual function, including glaucoma and ischemic optic neuropathy. Based on these results [34], it is suggested that the decreased expression of NeuN is associated with neuronal cell death at the retinal damage site [35]. As expected, ERCMs and PAGF-AB remarkably recovered the NeuN-positive RGCs which had been decreased by half in glaucoma eyes (Figure 4A,B).

Furthermore, the neuroprotective effects of ERCMs were confirmed via neuroglial cell activation since Müller glial cells are critically involved in the retinal inflammatory process, changing the biochemical microenvironment deteriorating the retina [36]. Increased IOP causes RGC stress and Müller cell activation by stimulating production of inflammatory cytokines such as TNF-α [37,38]. The immunoreactivities of GFAP and GS, the glial cell markers, significantly increased in Müller cells of the damaged area and inner and outer layers of the glaucoma retinas [39], which were markedly attenuated by treatment of ERCMs and PDGF-AB (Figure 4C,D). Although anti-inflammatory activities of MSC-derived exosomes have been reported [40,41], here we demonstrated anti-inflammation-mediated neuroprotective activities of ERCMs and PDGF-AB in retinopathy.

Neuroglial responses are well known to be involved in pathophysiology through inflammatory process, oxidative stress, and ischemic damage [42]. Reactive oxygen species and ultraviolet rays have been reported to induce cell death in the retina, including RPECs [43,44]. So, we checked the ERCM’s anti-oxidative and anti-ischemic potentials. Interestingly, AMMSC-ERCM (≥25 μg/mL) fully suppressed both the oxidative and hypoxic stresses of ARPE-19 cells and further increased the cell numbers (Figure 4E,F).

In the hyperosmotic glaucoma model, severe shrinkage of retinal tissues and inflammatory Müller cell activation were induced, resulting in IOP increase as well as retinal cell loss. However, ERCMs containing large amounts of GFs and NFs restored all the aspects of retinopathy, i.e., retinal shrinkage, Müller cell activation, and retinal cell death. Such in vivo beneficial effects ERCMs were supported by in vitro activities. That is, ERCMs containing GFs and NFs including PDGF facilitated proliferation of retinal cells and ameliorated their oxidative and hypoxic injuries. Notably, the exosomes in ERCMs were confirmed to penetrate RGCs. In our recent study, it was found that a large number of CM-DiI-labeled exosomes were observed in the eyes after intravenous injection, indicating that they can reach the injured retinal cells and tissues after local or systemic administration.

In conclusion, it was confirmed that AMMSC-ERCM and AMESC-ERCM containing large amounts of functional substances such as GFs and NFs, especially PDGF, protect retinal cells against oxidative and hypoxic damages in vitro and improve glaucoma by restoring intraocular pressure and retinal degeneration in vivo. Although follow-up studies to further clarify major functional molecules and underlying mechanisms, the stem cell ERCMs could be a candidate for the prevention and recovery of glaucoma at the early and middle phases of progress.

## 4. Materials and Methods

### 4.1. Preparation of Amniotic Membrane Stem Cells

AMMSCs and AMESCs were prepared under Good Manufacturing Practice conditions (Central Research Institute of Designed Cells Co., Ltd., Cheongju, Republic of Korea) as previously described [28]. In brief, the amniotic membrane tissues were digested with collagenase I, neutralized with an equal volume of medium containing 10% fetal bovine serum (FBS; Biowest, Kansas City, MO, USA), and centrifuged at 1500 rpm for 10 min. After washing twice, the contaminated red blood cells (RBCs) were lysed with RBC lysis buffer, and the remaining cells were suspended in Keratinocyte serum-free medium (SFM; Invitrogen, Carlsbad, CA, USA) supplemented with 5% FBS, 100 U/mL penicillin, and 100 mg/mL streptomycin (Invitrogen). Cultures were maintained under 5% CO_2_ at 37 °C in culture flask. Media were changed every 2–3 days.

The prepared amniotic stem cells were analyzed for their stem cell markers in a flow cytometric system. The AMMSCs and AMESCs were confirmed to be mesenchymal and epithelial stem cells, respectively [28].

### 4.2. Preparation of ERCMs

#### 4.2.1. Collection of ERCMs

The separated amniotic membrane stem cells were suspended in the defined serum-free medium in Hyper flask (Nunc, Rochester, NY, USA) and cultivated under normal oxygen (20% O_2_, 5% CO_2_) or hypoxic oxygen (2% O_2_, 5% CO_2_) tensions at 37 °C for 3 days. The media were filtered through a bottle-top vacuum filter system (0.22 μm, PES membrane) (Corning, Glendale, CA, USA). The conditioned media were 30-fold concentrated using Vivaflow-200 (Sartorius, Hannover, Germany).

#### 4.2.2. Western Blot Analysis of CD9

Protein-based quantification of CD9 of isolated exosomes was done using the protein DC assay kit (Bio-Rad Laboratories, Hercules, CA, USA). An aliquot (25 μg) of NCMs and ERCMs was denatured using ×6 denaturation buffer at 95 °C for 10 min and then resolved on 12% SDS-polyacrylamide gel by electrophoresis. Resolved proteins were transferred onto Immobilon-P PVDF membrane, and then blocked by incubating in 5% skim milk to minimize non-specific binding of antibodies. Blocked blots were submerged with primary antibodies for CD9 (1:1000; Abcam, Cambridge, UK) overnight at 4 °C, and subsequently washed three times (10 min each) with ×1 Tris buffer saline with 0.1% Tween-20 (TBS-T) buffer followed by incubation with HRP-conjugated secondary anti-mouse (1:2000; Abcam) antibodies for 2 h at room temperature. Unbound antibodies were removed by washing with ×1 TBS-T buffer (3 × 10 min), and signal was recorded using WestFemto maximum sensitivity substrate kit under Bio-Rad ChemiDoc Imager (Bio-Rad Laboratories), as described in Appendix A.

#### 4.2.3. NTA of Exosomes

The numbers of exosome nanoparticles in NCMs and ERCMs were analyzed using an NTA system (Nanosight NS300; NanoSight, Amesbury, UK) [45].

#### 4.2.4. Enzyme-Linked Immunosorbent Assay (ELISA) of GFs and NFs

The major GFs and NFs related to the retinal cell protection and regeneration in AMMSC-NCM and AMMSC-ERCM were determined via ELISA, according to the manufacturer’s instructions. Briefly, AMMSC-NCM or AMMSC-ERCM were put into the ELISA wells with antibodies specific for FGF (ab219636; Abcam), EGF (ab217772; Abcam), IGF (ab211652; Abcam), VEGF (ab222510; Abcam), TGF-β (ab100647; Abcam), BDNF (ab212166; Abcam), or PDGF (ab100622; Abcam) and incubated at room temperature for 0.5–1 h. After washing 3–4 times with wash buffer, the primary antibodies were added, and reacted at room temperature for 1 h and then washed again. Secondary antibodies were treated and incubated at room temperature for 30–45 min, washed, and substrates were applied to develop color for 10–30 min. After stopping the color development with a stop solution, the absorbance was measured at 450 nm.

### 4.3. Exosome-Uptake Assay in Retinal Cells

#### 4.3.1. Isolation and Culture of Rat RGCs

Five-week-old Sprague–Dawley (SD) rats (Daehan-Biolink, Eumsung, Republic of Korea) were sacrificed by lethal overdosage with a combination of ketamine and Rompun^®^ (Bayer Korea, Seoul, Republic of Korea). The retinal tissues were separated from the enucleated eyeballs of rats. Dissected retinas were enzymatically digested with 15 mL of 0.25% trypsin (Invitrogen, Carlsbad, CA, USA) for 30 min at 37 °C followed by trituration. Dissociated retinal cells were then sieved through a 40-μm nylon mesh (Millipore, Burlington, VT, USA) to remove tissue clumps, resulting in a retinal cell suspension, and centrifuged at 300× *g* for 5 min at 25 °C. The cell pellet was washed with Dulbecco’s Modified Eagle’s Medium (DMEM; Biowest) containing 10% FBS, 100 U/mL penicillin, and 100 μg/mL streptomycin. The cells were again centrifuged at 300× *g* for 5 min at room temperature and the supernatant was discarded [46,47,48].

pRGCs isolated from the retina were cultivated in DMEM/High glucose (Biowest) supplemented with 10% FBS, 100 U/mL penicillin, and 100 mg/mL streptomycin. The cultures were maintained under 5% CO_2_ at 37 °C in a culture flask. Media were changed every 2–3 days, and all experiments were conducted using the cells within the first 5 passages.

#### 4.3.2. Exosome-Uptake Assay

Exosomes in ERCMs obtained from AMMSCs were labeled with red CM-DiI membrane dye (C7000; Invitrogen) and prepared at 4 × 10^10^ particles/mL [45,49,50].

pRGCs were seeded on 8-well chamber slides (NUNC C7182; Thermo Fisher Scientific, Waltham, MA, USA) at 1 × 10^5^/mL. After 24 h, cells were damaged with 150 µM H_2_O_2_, and then incubated with labeled exosomes (50 μL/mL) for 4 h at 37 °C in a CO_2_ (5%) incubator. The cells were fixed with 4% paraformaldehyde for 20 min at room temperature. Fixed cells were treated with 0.1% Triton X-100 (Thermo Fisher Scientific) at room temperature. Five min later, they were washed in PBS, and blocked with 1% bovine serum albumin (BSA) for 1 h. The cells were immunostained with anti-α-tubulin antibody (1:1000, ab7291; Abcam, Cambridge, UK) for 2 h at 37 °C followed by goat anti-mouse IgG Alexa Fluor^TM^ 488 (1:500, Invitrogen) for 1 h at room temperature. The cell nuclei were stained with DAPI (Thermo Fisher Scientific) and examined under a microscope (BX51; Olympus, Tokyo, Japan) [45,49].

### 4.4. Retinal Cell-Proliferative and -Protective Activities of ERCM

#### 4.4.1. RPEC Culture

Human ARPE-19 cells were purchased from American Type Culture Collection (ATCC, Manassas, VA, USA). Cells were cultivated in Dulbecco’s Modified Eagle’s Medium/Nutrient Mixture F-12 (DMEM/F12; Biowest) supplemented with 10% FBS, 100 U/mL penicillin, and 100 mg/mL streptomycin. The cultures were maintained under 5% CO_2_ at 37 °C in a culture flask. Media were changed every 2–3 days, and all experiments were conducted using the cells within the first 5 passages.

#### 4.4.2. Cell-Proliferative Activity

In order to assess the cell-proliferating activity of ERCM, ARPE-19 cells (1 × 10^5^/mL) were seeded in a 96-well plate. The cells were treated with various concentrations (6.25–100 μL/mL) of AMMSC-ERCM. After 24 h culture in a normoxic condition (20% O_2_) at 37 °C, the cell number was counted.

#### 4.4.3. Cell-Protective Activities

To evaluate the cytoprotective activity of ERCM against oxidative stress, ARPE-19 cells (1 × 10^5^/mL) were seeded in a 96-well plate. The cells were exposed to 200 μM H_2_O_2_ and treated with AMMSC-ERCM (12.5–100 μL/mL). After 24 h culture in a normoxic condition (20% O_2_) at 37 °C, the cell number was counted.

To assess the cytoprotective activity of ERCM against hypoxic injury, ARPE-19 cells (1 × 10^5^/mL) were seeded in a 96-well plate. The cells were treated with various concentrations of ERCM (12.5–100 μg/mL). After 24 h culture in normoxic (20% O_2_) or hypoxic (2% O_2_) conditions at 37 °C, the cell number was counted.

### 4.5. Assessment of Glaucoma-Therapeutic Activity

#### 4.5.1. Animals

Six-week-old male SD rats were purchased from Deahan-Biolink. The animals were housed at a room with a constant temperature (23 ± 2 °C), relative humidity of 55 ± 10%, and 12 h light/dark cycle and fed with standard rodent chow and purified water ad libitum.

#### 4.5.2. Hyperosmotic Glaucoma Model and Treatment

Ocular hypertension was surgically induced in the left eye of 35 rats. Procedures were conducted under general anesthesia using a mixture of Zoletil^®^ (Virbac Korea, Seoul, Republic of Korea) and Rompun^®^ (Bayer Korea) at a ratio of 2:1, which was intraperitoneally injected at a volume of 2 mL/kg. A hypertonic (1.8 M) saline solution (50 μL) was injected into the two episcleral veins at a speed of 50 μL/min using a Hamilton microliter syringe (Hamilton, Reno, NV, USA) [51].

The IOP from both eyes of each rat was measured at regular intervals using a TonoLab tonometer (Icare Finland Oy, Vantaa, Finland) under inhalational anesthesia (0.4% isoflurane in oxygen). After a week of peak IOP, vehicle (0.9% saline, 10 μL/eye) or test materials, that is, AMMSC-NCM (8.8 × 10^6^ exosomes in 10 μL/eye), AMMSC-ERCM (4.5 × 10^8^ exosomes in 10 μL/eye), AMESC-ERCM (3.9 × 10^8^ exosomes in 10 μL/eye), or human PDGF-AB (1.5 μg in 10 μL/eye; 100-00AB; Peprotech, Cranbury, NJ, USA) were administered into the vitreous body. The IOP was measured at 10:00 a.m. twice a week until the end of the experiment.

#### 4.5.3. Retina-Preserving Activity

After sacrifice of the animals with deep anesthesia, the excised eyes were fixed in Davidson’s solution (BBC Biochemical, Mount Vernon, WA, USA) for 24 h. After fixation, washing, and dehydration, the tissues were embedded in paraffin, and cut into 4 μm sections using a microtome (Leica Biosystems, Ernst-Leitz-Straße, Wetzlar, Germany).

For general morphological examination, the slide sections of the eyes were stained with hematoxylin/eosin and observed with an optical microscope (Carl Zeiss, Jena, Germany). The thicknesses of retinal layers were measured, i.e., whole thickness, ganglion cell layer (GCL), inner nuclear layer (INL), and outer nuclear layer (ONL), as described in Appendix A.

For immunohistochemistry, the slides were incubated in 1% hydrogen peroxide for 30 min. After washing with PBS, a streptavidin-biotin peroxidase complex (LSAB2 kit; Dako, Carpinteria, CA, USA) was applied for 20 min and blocked in normal goat serum (Vectastain Elite ABC kit; Vector Laboratories, Burlingame, CA, USA) for 30 min. They were then incubated overnight at 4 °C with primary monoclonal rabbit antibodies against NeuN (1:1000; Abcam), GS (1:500; Abcam) or a polyclonal rabbit antibody against GFAP (1:500; Millipore, Burlington, VT, USA). The sections were treated with biotinylated secondary antibody (Vectastain Elite ABC kit; Vector Laboratories) at room temperature for 60 min. After washing with PBS, the tissues were color-developed with 3,3-diaminobenzidine tetrahydrochloride (DAB; Novus Biologicals, Centennial, CO, USA) for 1–2 min. All sections were counterstained with Harris hematoxylin, and examined focusing on the RGCs and Müller cells, as indicated in Appendix A.

### 4.6. Statistical Analysis

The results are presented as mean ± standard error. Statistical significance between the groups was determined by one-way analysis of variance followed by post hoc Tukey’s multiple comparison tests. *p*-values < 0.05 were considered statistically significant.

## Figures and Tables

**Figure 1 ijms-24-08073-f001:**
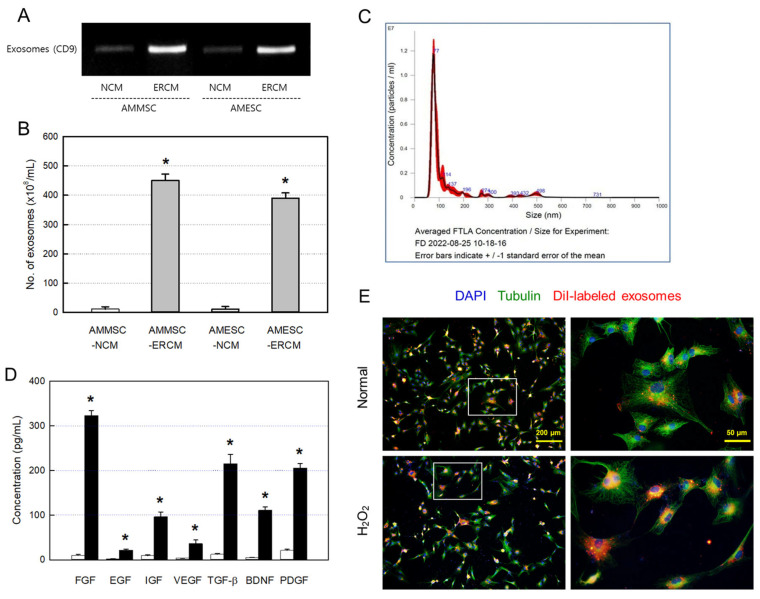
Exosomes obtained from amniotic membrane mesenchymal stem cells (AMMSCs) and amniotic membrane epithelial stem cells (AMESCs). (**A**) Contents of CD9-positive exosomes in normal conditioned medium (NCM) and exosome-rich conditioned medium (ERCM). (**B**) Number of exosomes in NCM and ERCM. (**C**) Particle size distribution of AMMSC exosomes. (**D**) Concentrations of growth factors and neurotrophic factors in NCM (white bars) and ERCM (black bars) of AMMSCs. (**E**) Penetration of CM-DiI-labeled AMMSC exosomes into primary retinal ganglion cells exposed to 150 µM H_2_O_2_ or its vehicle (Normal). FGF—fibroblast growth factor, EGF—epidermal growth factor, IGF—insulin-like growth factor, VEGF—vascular epithelial growth factor, TGF-β—transforming growth factor-β, BDNF—brain-derived neurotrophic factor, PDGF—platelet-derived growth factor. * Significantly different from each NCM (*p* < 0.05).

**Figure 2 ijms-24-08073-f002:**
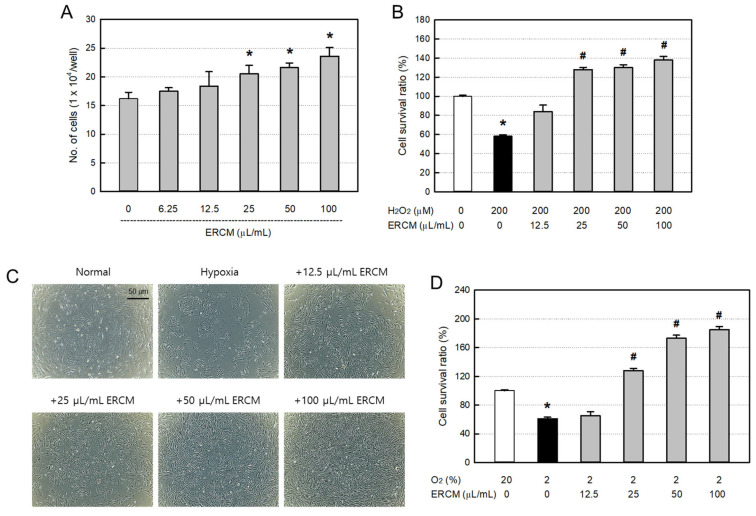
Retinal-cell-proliferative and -protective activities of exosome-rich conditioned medium (AMMSC-ERCM). (**A**) ARPE-19 cell proliferation by ERCM. (**B**) ARPE-19 cell protection by ERCM against oxidative stress (200 µM H_2_O_2_). * Significantly different from vehicle control (*p* < 0.05). ^#^ Significantly different from H_2_O_2_ alone (*p* < 0.05). (**C**) Representative findings of ARPE-19 cells exposed to hypoxic condition insult (2% O_2_) and treated with ERCM. (**D**) ARPE-19 cell protection by ERCM against hypoxic injury. * Significantly different from normoxic (20% O_2_) control (*p* < 0.05). ^#^ Significantly different from hypoxia (2% O_2_) alone (*p* < 0.05).

**Figure 3 ijms-24-08073-f003:**
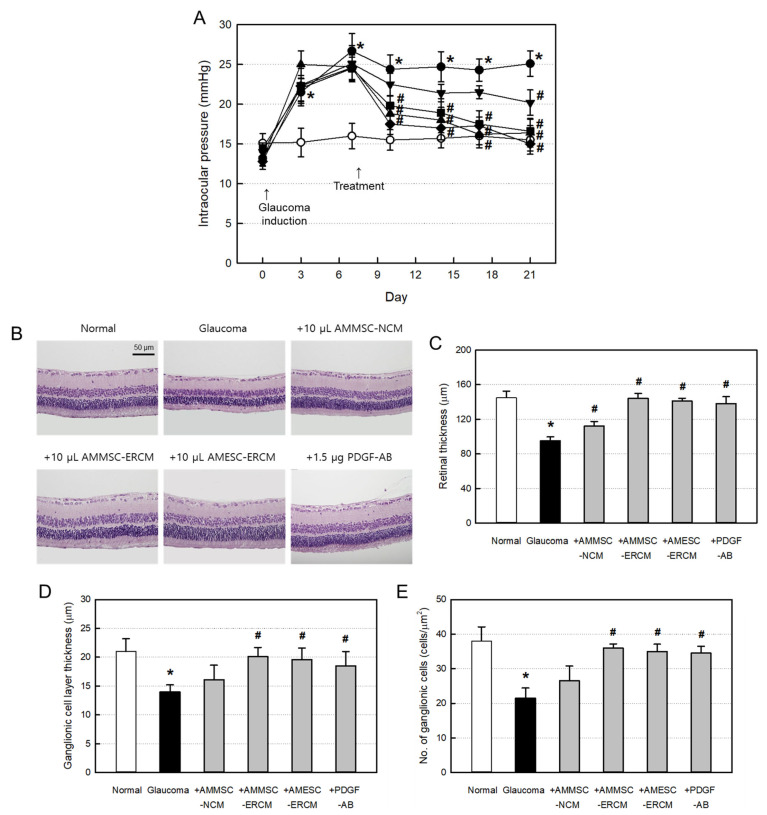
Intraocular-pressure (IOP)-lowering and retina-protective activities of exosome-rich conditioned media (ERCMs). (**A**) Change in IOP of glaucoma rats treated with normal conditioned medium (NCM), ERCMs (10 μL/eye) or platelet-derived growth factor-AB (PDGF-AB, 1.5 μg/eye). ○: Normal, ●: Glaucoma alone, ▼: Glaucoma + NCM, ■: Glaucoma + AMMSC-ERCM, ◆: Glaucoma + AMESC-ERCM, ▲: Glaucoma + PDGF-AB. (**B**) Representative findings of the retina of glaucoma rats. (**C**) Whole thickness of the retina. (**D**) Thickness of retinal ganglion cell layer. (**E**) Number of ganglion cells. * Significantly different from normal control (*p* < 0.05). ^#^ Significantly different from glaucoma alone (*p* < 0.05).

**Figure 4 ijms-24-08073-f004:**
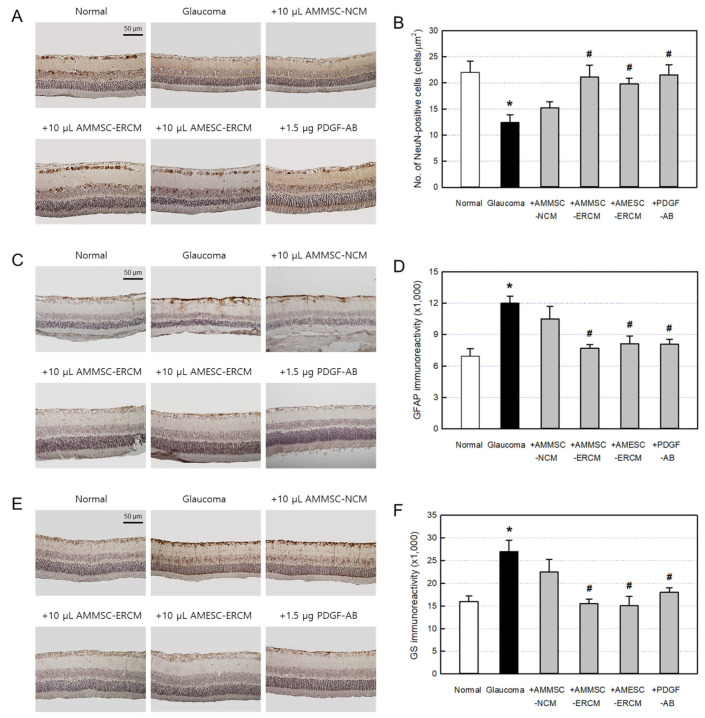
Retinal-ganglion-cell (RGC)-preserving and neuroprotective activities of exosome-rich conditioned media (ERCMs). (**A**,**B**) Representative findings and the number of NeuN-positive RGCs of glaucoma rats treated with normal conditioned medium (NCM), ERCMs (10 μL/eye) or platelet-derived growth factor-AB (PDGF-AB, 1.5 μg/eye). (**C**,**D**) Representative findings and the immunoreactivity of glial-fibrillary-acidic-protein (GFAP)-positive Müller cells. (**E**,**F**) Representative findings and the immunoreactivity of glutamine-synthetase (GS)-positive Müller cells. * Significantly different from normal control (*p* < 0.05). ^#^ Significantly different from glaucoma alone (*p* < 0.05).

## Data Availability

Data are contained within the article or Appendix A.

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
