# Peer review of "Intraocular Pressure-Lowering and Retina-Protective Effects of Exosome-Rich Conditioned Media from Human Amniotic Membrane Stem Cells in a Rat Model of Glaucoma"

_ijms, 2023, doi:10.3390/ijms24098073_

Round 1

Reviewer 1 Report

The authors showed that exosome-rich conditioned media had the protective effects in the glaucoma model. However, there are several questions should be addressed.

1. In ‘Materials and methods’, all the numbers before each subtitle should be changed to ‘4’.

2. In ‘Results’, there are 10 figures in total, it would be better to merge some figures together.

3. In ‘Results’, it would be better to add some data to support the exosome were delivered into the retina.

4. In figure 5, please add some statistical results in the figure. It would be better to change it into bar plot.

5. In figure 10, it would be better to add some other evidence to confirm the changes of GS, for example WB and PCR.

6. In ‘Results’, it would be better to add some electrophysiological test or behavioral test in vivo to confirm the effects of exosome.

7. In ‘Results’, it would be better to add some data about the mechanism why the exosome contribute to the improvement of glaucoma rats.

Author Response

Answers to reviewers’ comments

Reviewer 1:

The authors showed that exosome-rich conditioned media had the protective effects in the glaucoma model. However, there are several questions should be addressed.

1. In ‘Materials and methods’, all the numbers before each subtitle should be changed to ‘4’.

→ Thank you for your correction!

2. In ‘Results’, there are 10 figures in total, it would be better to merge some figures together.

→ We merged to 4 figures and rearranged the figures. Thank you!

3. In ‘Results’, it would be better to add some data to support the exosome were delivered into the retina.

→ Thank you for your valuable advice! We are planning to investigate the exosomes’ in vivo penetration into the retina. It is expected to report the results in a follow-up study in several months.

4. In figure 5, please add some statistical results in the figure. It would be better to change it into bar plot.

→ We added. Thank you!

5. In figure 10, it would be better to add some other evidence to confirm the changes of GS, for example WB and PCR.

→ I am sorry that we have no enough sample for WB and PCR analyses. We are conducting a follow-up study to obtain additional data, although it may take several months.

6. In ‘Results’, it would be better to add some electrophysiological test or behavioral test in vivo to confirm the effects of exosome.

→ Actually, the animals were sacrificed. So, we are keeping in mind to obtain more profound data, including vision acuity recovery in Shuttle box discrimination performances, as you recommended. Thank you!

7. In ‘Results’, it would be better to add some data about the mechanism why the exosomes contribute to the improvement of glaucoma rats.

→ Thank you for your valuable advice! In previous studies, restoration of trabecular meshwork was suggested as an IOP-lowering effect of PDGF, an active ingredient in stem cells’ conditioned medium. We are planning to investigate the restoration of trabecular meshwork via electron microscopy. However, the experiments are time-consuming. So, we can report the results in a next submission. Retinal cell-proliferative and -protective activities of exosomes under oxidative and hypoxic conditions were confirmed in the present study.

Reviewer 2 Report

The introduction part can be improved by including the mechanism of action, safety, and efficacy of exosomes based on previous research studies in the treatment of glaucoma.

Author Response

Answers to reviewers’ comments

Reviewer 2:

Comments and Suggestions for Authors

The introduction part can be improved by including the mechanism of action, safety, and efficacy of exosomes based on previous research studies in the treatment of glaucoma.

→ We added. Please see the Introduction and Discussion sections. Thank you!

Reviewer 3 Report

In this manuscript, Hye-Rim Seong et al. showed that AMMSC-ERCM and AMESC-ERCM containing large amounts of functional substances such as GFs and NFs, especially PDGF, protect retinal cells against oxidative and hypoxic damages in vitro and improve glaucoma by restoring intraocular pressure and retinal degeneration in vivo.

There are no major problems with this manuscript; the data are very well presented in text, pictures and graphs.

Here are my comments and suggestions

1.        Describtion of ELISA test is too brief. Informations about used antibodies should be added.

2.        The body of Section 2.2.2 is not related with the title of the section.

3.        Morphological pictures should be extended with the names of each regions of rat retina and identified factor.

4.        In line 252 the phrase:  treated with,  is reapeted.

5.        However, p-values were shown, individual analyzes were not numbered.

Author Response

Answers to reviewers’ comments

Reviewer 3:

In this manuscript, Hye-Rim Seong et al. showed that AMMSC-ERCM and AMESC-ERCM containing large amounts of functional substances such as GFs and NFs, especially PDGF, protect retinal cells against oxidative and hypoxic damages in vitro and improve glaucoma by restoring intraocular pressure and retinal degeneration in vivo.

There are no major problems with this manuscript; the data are very well presented in text, pictures and graphs.

Here are my comments and suggestions

1. Describtion of ELISA test is too brief. Informations about used antibodies should be added.

→ We described the ELISA test in more detail including information on antibodies. Thank you!

3. The body of Section 2.2.2 is not related with the title of the section.

→ Sorry for disturbing you! We corrected RT-PCR to WB of CD9. Thank you!

3. Morphological pictures should be extended with the names of each regions of rat retina and identified factor.

→ We added magnified images and arrows for the identified factors in Supplementary figures. Thank you!

4. In line 252 the phrase: treated with, is repeated.

→ We corrected. Thank you!

5. However, p-values were shown, individual analyzes were not numbered.

→ Thank you for your advice! But, we analyzed the significances between Control vs Glaucoma and between Glaucoma vs Treatment groups.

Round 2

Reviewer 1 Report

All the comments are responded, I agree to accept it.